# Assessment of Local People Opinion After World Heritage Site Designation, Case Study: Historic City of Yazd, Iran

**Ahmad Nasrolahi** [1],*[ID], **Jean-Michel Roux** [2], **Leila Ghasvarian Jahromi** [3] **and Mahmoudreza Khalili** [4]

[1] Department of International Development in Urban Planning (ICUP), Grenoble University Alpes, 14 av. Mr Reynoard 38100 GRENOBLE, France

[2] Urban Planning and Alpine Geography Institute, Grenoble University Alpes, 14 av. Mr Reynoard 38100 GRENOBLE, France; jean-michel.roux@univ-grenoble-alpes.fr

[3] Department of Conservation, Arianam Company, No. 29, First East Alley, North Hafez Ave., Shahinshahr 81464, Iran; leilajahromi@gmail.com

[4] Department of Urban and Regional Planning, Technical University of Berlin, Straße des 17. Juni 135, 10623 Berlin, Germany; khalili.mahmoudreza@gmail.com

* Correspondence: ahmadnasrolahi@gmail.com; Tel.: +49-1767-433-8187

**Abstract:** Local participation in the cultural heritage conservation has always been a concern since the Venice Charter (1964). It seems the assumption of the World Heritage Center, and particularly their State Parties, is that local people living in a nominated site are willing to inscribe their properties on the World Heritage List. This research examines the points of view of a local community living in the buffer zone of the Historic City of Yazd in five categories: Willingness, quality of life, decision-making, benefits, and awareness after the designation as World Heritage Site. The main hypotheses are that local people did not agree to inscribe their properties on the World Heritage List, and their quality of life has not changed after registering. The methodology is based on both qualitative and quantitative methods by interviewing 400 people of both genders and different ages. The results show that the majority of local people living in the buffer zone were not satisfied to be on the list. In addition, more than 80% mentioned that the quality of life did not change at all after the inscription. There was a misunderstanding about the role of national and international organizations in World Heritage management and conservation among the local community.

**Keywords:** local people participation; cultural heritage conservation; World Heritage Site; Historic City of Yazd; decision-making

## 1. Introduction

Community participation in cultural heritage conservation has been a concern ever since the Venice Charter (1964) and is still to this day [1]. This approach has also been highlighted in World Heritage Documents. In addition, the Faro Convention (2005) adopted to shift the focus from the conservation of cultural heritage values to the value of cultural heritage for the society. In this case, it is necessary to engage local people participation in all stages of cultural heritage conservation and management [2]. Moreover, a number of papers have focused on the importance of public participation in heritage conservation and tourism management [3,4].

Moreover, the Dresden Elbe Valley example opens a debate about whether local people would be willing to live in a World Heritage Site or not if it was up for a vote. The construction of the Waldschlößchenbrücke Bridge was vital for the city, which led the Federal Republic of Germany to put

the decision up for locals to vote whether they wanted a construction of the Bridge (which meant being delisted), or being on the World Heritage List. Interestingly, a little over half of the eligible people participated in the referendum with 67.92% voting for the first option [5].

Everyone knows that living in World Heritage Sites (e.g., historic cities) is not the same as living in public or governmental buildings or sites where local people participation is able to enhance the conservation and management process. Historic cities are a place of everyday life and a place for the activities of their local people, generation after generation. They belong to the people. In fact, World Heritage Site designation affects all aspects of a local community's life in these areas [6]. This paper examines the views of local people living in Yazd, a World Heritage Site, in five categories: Willingness, quality of life, decision-making, benefits, and awareness.

The Historic City of Yazd World Heritage Site is located in the middle of the central plateau of Iran, 621 km southeast of Tehran. "The nominated property consists of three components covering an area of 195.76 ha and includes the historic city center, the Zoroastrian district, and the Dolat-abad Persian garden, which is also a component of the serial World Heritage property . . . . The buffer zone encompasses the three nominated components and covers an area of 665.93 ha." The city "has about sixty districts. Nineteen districts are located within the Historic City of Yazd. Districts are characterized by professional, ethnic or religious concentrations" [7]. Around one third of the urban area is located in the historical city. More than 436,000 people (around 50,000 households) are living in the urban area [8] (Figure 1). This area, now known as Historic City of Yazd, was inscribed on the World Heritage List in 2017.

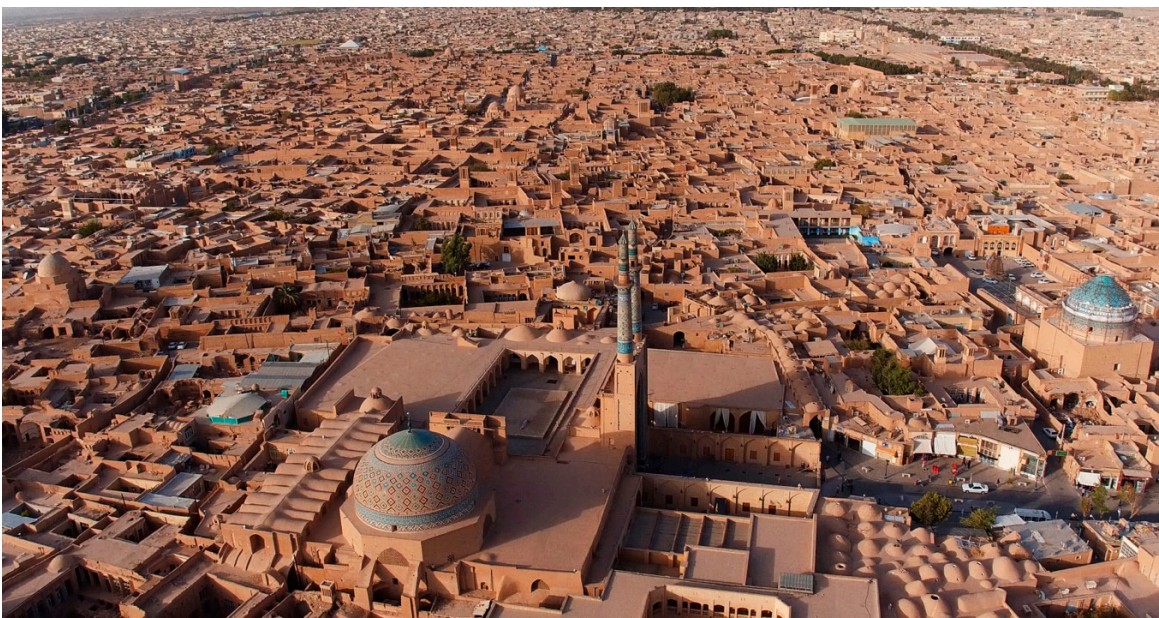

**Figure 1.** Historic City of Yazd, aerial view, Jame Mosque [9].

Before inscribing on the World Heritage List, local people were happy that the city was going to be a part of world heritage, but after listing, concerns arose about what the benefits were for living in a World Heritage Site [10]. It seems that, for some State Parties, the inscription of a property on the World Heritage List is not to improve cultural heritage conservation, protection, and providing advantages for the local community, but is done for the sake of modifying international prestige and being under the flag of the UNESCO title.

Besides, although participatory approach is highlighted in the World Heritage Documents, there are no indicators for an assessment of local people participation in the nomination dossier. It seems that local people opinion is not that important for the World Heritage Center and they just emphasize on the State Parties' decision, because all State Parties are responsible in all steps (e.g., providing the

Tentative List, nomination dossier, annual report, etc.) from nominating a property to the management of World Heritage Sites. Local people who are the main stakeholders, the real owners of the cultural and World Heritage Site, are deliberately or unintentionally ignored by some State Parties.

According to the World Heritage Convention, the process of inscription of a property on the World Heritage List is to prepare a nomination dossier by the State Party. According to the Resource Manuals by the World Heritage Centre, "the first step a given State Party must take is to make an 'inventory' of its important natural and cultural heritage sites located within its boundaries. This 'inventory' is known as the Tentative List and provides a forecast of the properties that a State Party may decide to submit for inscription in the next five to ten years and which may be updated at any time. It is an important step since the World Heritage Committee cannot consider a nomination for inscription on the World Heritage List unless the property has already been included on the State Party's Tentative List."

"By preparing a Tentative List and selecting sites from it, a State Party can plan when to present a nomination file. The World Heritage Centre offers advice and assistance to the State Party in preparing this file, which needs to be as exhaustive as possible, making sure the necessary documentation and maps are included. The nomination is submitted to the World Heritage Centre for review and to check it is complete. Once a nomination file is complete the World Heritage Centre sends it to the appropriate Advisory Bodies for evaluation" [11].

Then, "A nominated property is independently evaluated by two Advisory Bodies mandated by the World Heritage Convention; The International Council on Monuments and Sites (ICOMOS) and the International Union for Conservation of Nature (IUCN), which respectively provide the World Heritage Committee with evaluations of the cultural and natural sites nominated. The third Advisory Body is the International Centre for the Study of the Preservation and Restoration of Cultural Property (ICCROM), an intergovernmental organization which provides the Committee with expert advice on conservation of cultural sites, as well as on training activities. Once a site has been nominated and evaluated, it is up to the intergovernmental World Heritage Committee to make the final decision on its inscription. Once a year, the Committee meets to decide which sites will be inscribed on the World Heritage List. It can also defer its decision and request further information on sites from the State Parties" [12].

In this case, the main question is whether local people who are living in the World Heritage Zone (as well as the buffer zone) are satisfied to live in a World Heritage Site or not. In addition, before nominating the Historic City of Yazd for inscription on the World Heritage List, were local people asked about their opinions on the matter? Furthermore, has the quality of life modified after inscribing on the World Heritage List?

## 2. Materials and Methods

Although it is necessary to engage all main-participant sectors (private, public, and civil society) in a participatory approach in urban context as potential partnerships [13], this paper focuses on city consultation as the property owners in the Historic City of Yazd World Heritage Site. In this research, three variables were examined by a questionnaire of 23 questions and 400 interviewees during a period of around two months between 2 March and 25 April, 2019. The variables were local people satisfaction, knowledge of local people in local, national and international organization responsibilities, and local people rights in cultural heritage conservation. In this case, three sorts of questionnaires were designed for the interview. The first category was about the satisfaction of the local people living in the buffer zone of the historic city before and after registering for the World Heritage List. The second category was the role of local people in the decision-making process, and the last category was the role of the Iranian Cultural Heritage Organization and the World Heritage Center in conservation and management.

The population of Yazd living in the urban area is 436,742. According to Social Scientific tools, if the confidence level and the confidence interval are 95% and 5%, respectively, the number of questionnaires must be for more than 384 interviews, which is acceptable in this particular research. Confidence level (also called margin of error) shows "the plus-or-minus figure usually reported in

newspaper or television opinion poll results." Confidence interval displays how certain the survey can be, "the 95% confidence level means the research can be 95% certain" [14] (Figure 2).

**Figure 2.** Determine sample size tool.

Of course, designed questionnaires follow its basic regulations. The questions "are as short as possible, are not leading or have implicit assumptions, do not include two questions in one, only exceptionally invite yes/no answers, are not too vague or general, do not use double negatives, are not, in any sense, invasive, or asking questions that the respondent is unlikely to want to answer, and do not invite respondents to breach confidentiality" [15]. Moreover, interviews must include all possible ages.

Since the main concept of participation was maximum engagement, the research was designed to include all people over 15 years of age in the interview. It was obvious that participants younger than 15 years required a specific sort of questionnaire which was redundant in this survey. In the Historic City of Yazd, the population census of 2015 showed that 436,742 people were living in the urban area, which were divided into age groups for this research: Five groups with a 10 year range for each group, and one group for participants older than 65 years (Table 1).

**Table 1.** The number of different genders in different age groups.

| Gender | Group A 15–24 | Group B 25–34 | Group C 35–44 | Group D 45–54 | Group E 55–64 | Group F >65 | Total |
|---|---|---|---|---|---|---|---|
| Woman | 41,307 | 63,926 | 45,357 | 30,098 | 19,282 | 15,447 | 215,417 |
| Man | 40,949 | 62,559 | 47,712 | 32,546 | 21,988 | 15,571 | 221,325 |

In addition, in order to apply gender equality, the exact number of women and men in each group was calculated. For example, the population census displayed that 41,307 females between the age of 15 to 24 (Group A) were living in the urban area, which were equivalent to around 19% of all females, and also 40,949 males (18%) were reported in this group. The more populated group for both genders was Group B, which covered around 30% and 28% of all women and men, respectively. The smallest group was Group F where the percentage of women and men was equal (7%) (Table 2).

**Table 2.** The percentage of women and men in each age group.

| Gender | Group A 15–24 | Group B 25–34 | Group C 35–44 | Group D 45–54 | Group E 55–64 | Group F >65 | Total |
|---|---|---|---|---|---|---|---|
| Women | 19% | 30% | 21% | 14% | 9% | 7% | 100% |
| Men | 18% | 28% | 22% | 15% | 10% | 7% | 100% |

The percentage of the population aimed for the research to justly distribute the questionnaires among all groups and also follow gender equality. Thus, the research needed to interview 38 women and 36 men between 15 and 24 (Group A). In order to have a correct evaluation, the number of interviewees older than 65 was not allowed to exceed 7% in both genders. The final step was to equally distribute the questionnaires in the entire core and buffer zone (Table 3).

**Table 3.** The number of interviewees based on gender and age group.

| Gender | Group A 15–24 | Group B 25–34 | Group C 35–44 | Group D 45–54 | Group E 55–64 | Group F >65 | Total |
|--------|------|------|------|------|------|------|-------|
| Women | 38 | 60 | 42 | 28 | 18 | 14 | 200 |
| Men | 36 | 56 | 44 | 30 | 20 | 14 | 200 |

According to Corrado Minervini, the method to fairly ask local people about their attitude in urban areas is equivalent reticulation, which means gridding the buffer zone for distributing 400 questionnaires [16]. The size of gridding depends on the size of the area. In this case, the gridding points sometimes were matched on the buildings. Otherwise, the nearest buildings were interviewed (Figure 3).

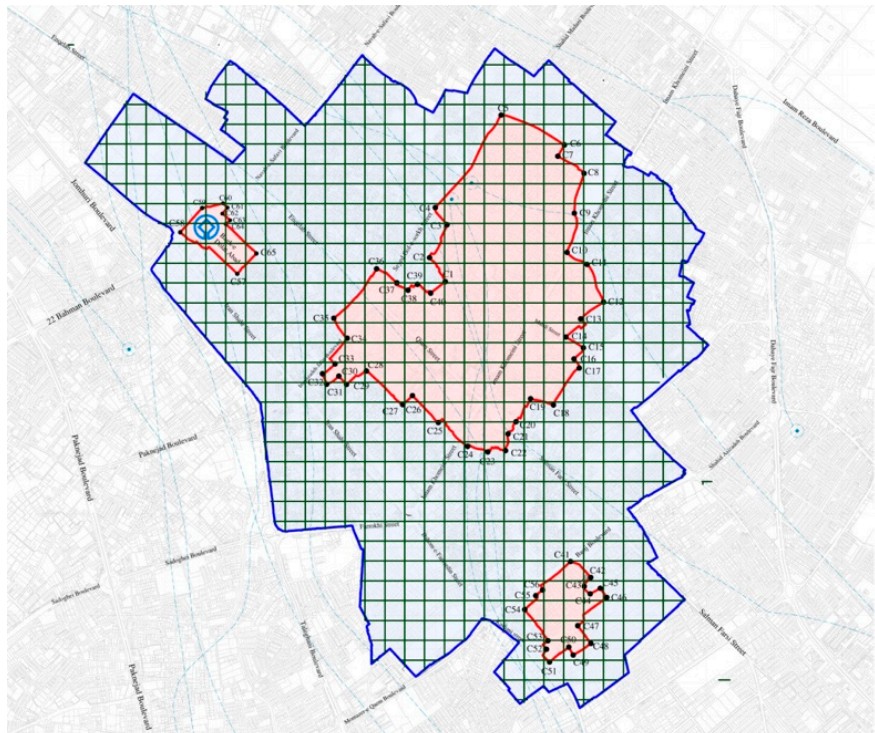

**Figure 3.** Gridding of the Historic City of Yazd buffer zone.

The questionnaire included general questions related to income, education, age, and gender. The rest of the questions were about the main research hypotheses: Local people satisfaction of living in a World Heritage Site, the role of local people in the decision-making process, and the knowledge of the local people of the responsibilities of national and international organizations in regards to heritage management and protection. This paper focuses on the results of five main questions concerning the local people agreement, the change of the quality of life, the decision-making process, benefits of inscribing on the World Heritage List, and the responsibilities of the organizations.

## 3. Results

### 3.1. General Economic Findings

According to the general questions, twenty-five percent mentioned that their monthly income was between 400 and 600€, between 600 and 800€ for another twenty-five percent, and more than 800€ for yet another twenty-five percent. Sixteen percent of interviewees earned less than 200€ and just nine percent said their income was between 200 and 400€. The majority of interviewees (75%) remarked

that their income was not related to the tourism industry at all. More than five percent said that less than one fourth of their income came from tourism. Interestingly, only 66 interviewees (out of 400) said that more than seventy percent of their income depended on the tourism industry. In the case of improving the economic situation after registering for the list, around fifty-five percent commented that it was going to worsen the economic situation, and more than forty-five percent said that it would not make a difference to them.

## 3.2. Local People Willingness

All interviewees mentioned that they were not asked about whether or not they would like to inscribe their properties onto the World Heritage List, and more than sixty-five percent expressed directly that they did not agree to be on the list. Furthermore, twenty-seven percent of local people have displayed that their viewpoints were not important to the government in this case. Only five percent agreed with the decision before and after inscribing on the list (Figure 4).

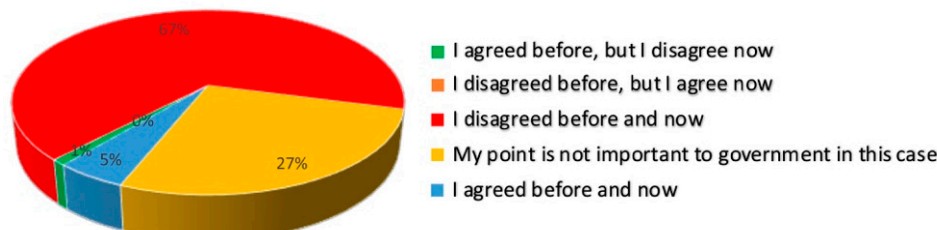

**Figure 4.** Local people opinion about inscription of Historic City of Yazd on the World Heritage List.

A majority of interviewees mentioned that the quality of life has not changed after recognizing the Historic City of Yazd as a World Heritage Site. On the contrary, sixteen percent had a different opinion and said that the quality of life has improved after inscription (Figure 5).

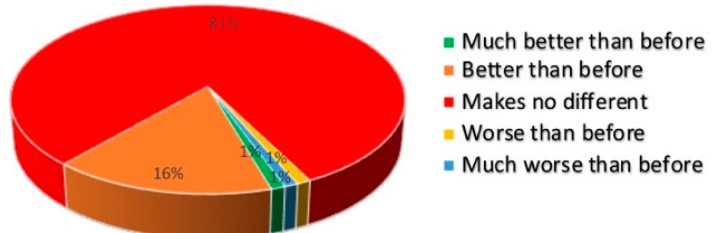

**Figure 5.** Local people opinion about improved quality of life after inscription on the World Heritage List.

## 3.3. The Role of Local People in Decision-Making

The role of local people in the decision-making process was asked in the questionnaire. More than seventy percent said that the World Heritage Center and the UNESCO were responsible for the decision-making process. The rest mentioned that the Iranian Cultural Heritage Organization and preservationists had to make the decision for them (Figure 6).

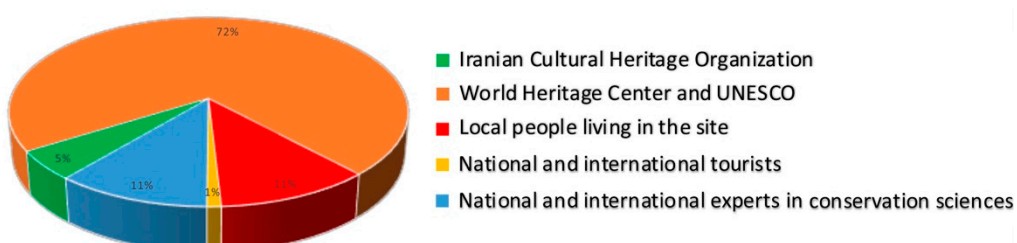

**Figure 6.** Local people opinion about who made the decision for the Historic City of Yazd.

In another section, the question was who the main beneficiary of the World Heritage Site was. It was surprising that only four percent believed that local people living in the site were the main beneficiaries. Fifty-two percent remarked that hotels and other tourist-related jobs, and other businesses and markets benefited from being on the site. Seventeen percent said that national and international tourists were taking advantage of World Heritage Sites and the rest of the interviewees (27%) commented that the local and national government, particularly the Iranian Cultural Heritage Organization, were the main beneficiaries of the World Heritage Site (Figure 7).

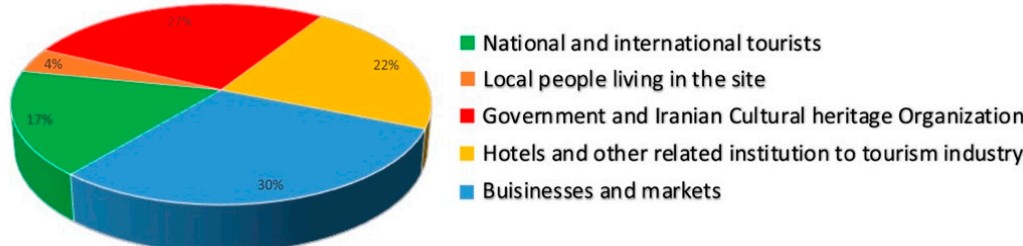

**Figure 7.** Local people opinion about who benefits most from World Heritage Sites.

*3.4. Local People Knowledge of the Responsibilities of National and International Organizations*

It is impressive to mention that twenty-four percent of the interviewees recognized the World Heritage Center and the UNESCO as the ones who were responsible for heritage protection. More than one fifth said that the local and national government were responsible for that. More than forty percent expressed that the government with local people assistance could preserve the site. Only a little over one-tenth were aware that local people living in the site with government assistance were able to protect heritage sites (Figure 8).

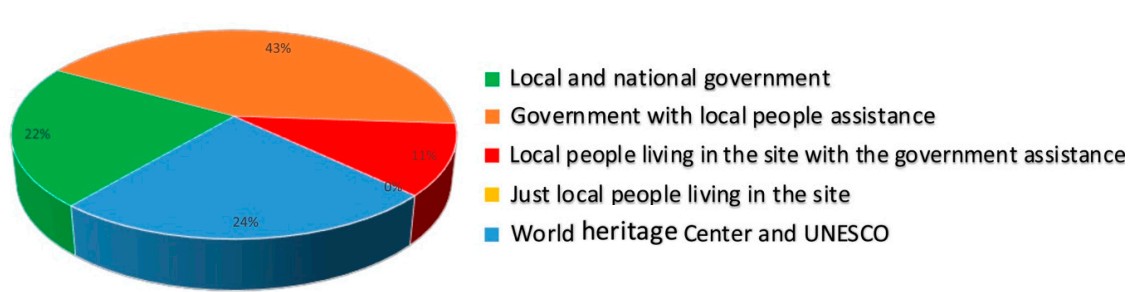

**Figure 8.** Local people opinion about who can preserve the Historic City of Yazd World Heritage Site.

## 4. Discussion

*4.1. Lessons Learnt from the Case Study*

According to Article 27 of the World Heritage Convention, one of the most important objectives of the Convention is "to increase the participation of local and national population in the protection and presentation of heritage" in order to encourage support for the World Heritage Convention, (also Article 5a). In addition, in 2017, according to the 39th session of the committee, participation of local, indigenous people, governmental, non-governmental and private organizations, and other stakeholders in conservation of World Heritage Properties, is necessary to share the responsibility with the State Party. All State Parties are encouraged to prepare a nomination dossier with the widest possible participation of stakeholders to secure prior and informed consent of indigenous people [17]. However, what happened in practice after the inscription on the list in Iran is controversial.

In fact, State Parties are responsible for all stages in the nomination process. They decide "which properties to include in the Tentative List", when and which properties in the Tentative List are to

be nominated for World Heritage Listing, and, furthermore, they "are responsible for the continuing protection and effective management of the property to the requirement of the World Heritage Convention" [18]. So, the question that arises here is: What is the role of local inhabitants?

In general, according to the viewpoint of local people living in the buffer zone, the results of the interviews were:

- The majority of the local people were not satisfied living in the World Heritage Site;
- The quality of life did not improve after the inscription of the city on the World Heritage List;
- Local people were excluded from the decision-making process of the nomination;
- Local people did not benefit from being the main stakeholders;
- There was a misunderstanding of the responsibilities of local people, government, and international organizations toward World Heritage Conservation among local people.

### 4.2. Need for Clear Definitions and Structures

Here it should be noted that there is no indicator to demonstrate how the World Heritage Committee evaluates indigenous participation. Although some parts of the nomination dossier include local participation, the measurement system of the World Heritage Committee for measuring the form of participation is not clear. Furthermore, in the Resource Manuals, it is unclear to what extent local people must participate. If local people are only being informed in this respect, it is exactly the lowest level of participation, according to the "Ladder of Citizen Participation" [19]. Finally, the role of locals in the conservation and presentation of a cultural heritage inscribed on the World Heritage list is unclear.

Furthermore, local people basically do not know that much about what exactly World Heritage Properties mean. In fact, the idea of inscription of a given property in a global agenda is very attractive. Increasing the number of national and international tourists, improving financial resources, involving international cooperation in conservation, etc., are the things that local people understand about the World Heritage Listing. But this glamorous, deceptive idea has a number of hidden losses for locals, at least in Iran. The best example in this case is the "Dresden Elbe Valley" that inscribed on the World Heritage List in 2004 and delisted in 2009. So, if local people understood the consequences of an inscription on the World Heritage List, most often, they would not like to be on the list.

## 5. Conclusions

The system of the World Heritage Nomination is a completely top-down decision-making process in Iran. There are long Tentative Lists provided by the Iranian Cultural Heritage, Handicraft, and Tourism Organization which is officially the very governmental organization. Besides, every year the Bureau of World Heritage Nomination, which is also officially the Iranian Cultural Heritage, Handicraft, and Tourism Organization, decides which property in the Tentative List should be nominated for listing. Although the World Heritage Center persists in local participation in the nomination process, this process is still top-down without any local people agreement. Therefore, there is no local participation in the process. Local people living in the Historic City of Yazd are the real owners of the site, but no one asked them whether they would like to inscribe their properties on the World Heritage List or not. In addition, the research shows that the inscription of the Historic City of Yazd on the World Heritage List did not lead to an improvement in the inhabitants' quality of life which could act as an incentive. Even by considering the growth in the tourism sector and a stricter protection of the cultural heritage by the Iranian Cultural Heritage Organization, they face serious problems in the buffer zone.

In general, if the World Heritage Center adopted a way to apply participatory approach in the nominating process, State Parties would oblige to involve local communities in heritage management by implementing different activities, such as holding training workshops for local people and modifying national rules and regulations related to effective participation in order to engage local people in decision-making and dividing the benefits of the World Heritage Listing among them.

**Author Contributions:** Project administration, writing—review and editing, A.N.; data collection and software, L.G.J.; data analysis, writing—review and editing, M.K.; supervision, J-M.R.

**Funding:** This research received no external funding.

**Acknowledgments:** We would also like to show our gratitude to Corrado Minervini for sharing his knowledge in data collection and methodology.

**Conflicts of Interest:** The authors declare no conflict of interest.

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
