# Peer review of "Assessment of Local People Opinion After World Heritage Site Designation, Case Study: Historic City of Yazd, Iran"

_heritage, doi:10.3390/heritage2020106_

Round 1

Reviewer 1 Report

This work deals with a case study: the inscription  of the  Historic City of Yazd, Iran, in the UNESCO  World Heritage List, focusing on the specific process of this Iranian  property. This process has been carried out in a top-down way and not bottom-up. Therefore the authors examine the lack of involvement of  common population in the inscription process and how people have not judged this process positively. This paper seems rather a complaint ...

The methodology of analysis (interviews) and data processing are correct and exhaustive. However,  in my opinion, in the "Conclusions"  it would be correct to point out that at the time of the interview little time has passed since the inscription of the City in the list (2017) to be able people to see the social-economic benefits and better explain the lack of knowledge about cultural property in this specific area of Iran.

Besides, to be published as case study or technical report, it is necessary to review the English language and to make some revisions:

- line 102, specify the exact period of questionnaire not only the time frame;

- lines 147-156, the same sentence is repeated twice;

- line 182, the  caption of figure 3 should be moved upwards.

Author Response

Thank you very much indeed for your feedback. It is my appreciate to have your comments on our research. I have modified all points you mentioned. Please find the attached file.

Best Regards,

Ahmad Nasrolahi  

Reviewer 2 Report

Although the research theme can be considered interesting, all the work was let down by the writing-up process and lack of coherent structure of ideas and data. The work was hindered by a large amount of direct citations and a very basic selection of references. Unfortunately, there are mistakes and inconsistences in terms of methodology and inaccuracy of references. Furthermore, the discussion of the results is poor, and the article is greatly hindered by major use of direct citations.

Author Response

Thank you very much indeed for your feedback. It is my appreciate to have your comments on our research. We have modified all points you mentioned including; lack of coherent structure, direct citations, referencing and results. 

Best Regards,

Ahmad Nasrolahi  

Reviewer 3 Report

I would suggest not to put 'Historic City' twice in the intro. Or maybe rework title to have a shorter/catchier title

This is a good start but the work needs to be expanded and offer a more critically informed discussion of the results. Perhaps also display results as a table opposed to presenting as is by question. Give more depth in analysis in your literature review as well and when interpreting your results. Your methods is explained. I suggest you referencing the following sources to get you started with this study to develop your academic contribution more:

Jimura, T. (2011). The impact of world heritage site designation on local communities–A case study of Ogimachi, Shirakawa-mura, Japan. Tourism Management32(2), 288-296.

Jimura, T. (2016). World heritage site management: a case study of sacred sites and pilgrimage routes in the Kii mountain range, Japan. Journal of Heritage Tourism11(4), 382-394.

Jimura, T. (2018). World Heritage Sites: Tourism, Local Communities and Conservation Activities. CABI.

Another paper on local perceptions at a potential world heritage site, see link: https://journals.sagepub.com/doi/abs/10.1177/0269094217734329?journalCode=leca

Work by Dallen Timothy as well on Heritage

Author Response

Thank you very much indeed for your useful feedback. It is my appreciate to have your comments on our research. We have modified all points you mentioned. 

Best Regards,

Ahmad Nasrolahi  

Reviewer 4 Report

This paper analyses an interesting topic. Nowadays the role of heritage as a cultural resource is an important field of work in economic research. I congratulate the authors for their work.

In my opinion this paper can be improved in two ways.

- First: Further development of the theoretical framework, the state of the art. It is necessary know if there are studies of the same type.

- Second:expand the conclusions. Causes? Consequences? Solutions?

Author Response

Thank you very much indeed for your feedback. It is my appreciate to have your comments on our research. We have modified all points you mentioned.

Best Regards,

Ahmad Nasrolahi  

Round 2

Reviewer 2 Report

After the review and improvement of the article, by the authors, the article is publishable.

Reviewer 3 Report

I am happy with the revisions made by the authors on the revised version of the manuscript